# Remote evaluation of STH program coverage: Experiences from the DeWorm3 study, India

**Kumudha Aruldas[1‡], Rohan Michael Ramesh[1‡], William E. Oswald[2,3], Venkateshprabhu Janagaraj[1], Angelin Titus[1], Jabaselvi Johnson[1], Malvika Saxena[1], Gideon John Israel[1], Katherine Halliday[2], Judd L. Walson[4,5], Arianna Rubin Means[5,6], Sitara Swarna Rao Ajjampur🄳[1]***

**1** The Wellcome Trust Research Laboratory, Division of Gastrointestinal Sciences, Christian Medical College, Vellore, India, **2** Department of Disease Control, Faculty of Infectious and Tropical Diseases, London School of Hygiene & Tropical Medicine, London, United Kingdom, **3** Global Health Division, International Development Group, Research Triangle Institute International, Research Triangle Park, North Carolina, United States of America, **4** Departments of Global Health, Medicine, Pediatrics and Epidemiology, University of Washington, Seattle, Washington, United States of America, **5** The DeWorm3 Project, University of Washington, Seattle, Washington, United States of America, **6** Department of Global Health, University of Washington, Seattle, Washington, United States of America

‡ These authors are co-lead authors.
* sitararao@cmcvellore.ac.in

**Data Availability Statement:** All relevant data are within the manuscript and its supporting information files

## Abstract

### Background

The DeWorm3 trial is a multi-country study testing the feasibility of interrupting transmission of soil-transmitted helminths by community-wide mass drug administration (cMDA). Treatment coverage during cMDA delivery was validated by in-person coverage evaluation surveys (CES) after each round of treatment. A mobile phone-based CES was carried out in India when access to households was restricted during the COVID-19 lockdown.

### Methods

Two focus group discussions were conducted with the survey implementers to document their experiences of conducting phone-based CES via mobile-phone voice calls.

### Principal findings

In the phone-based CES, only 56% of sampled households were reached compared to 89% during the in-person CES (89%). This was due to phone numbers being wrongly recorded, or calls being unanswered leading to a higher number of households that had to be sampled in order to achieve the sample size of 2,000 households in phone-based CES compared in-person CES (3,600 and 2,352 respectively). Although the phone-based CES took less time to complete than in person coverage evaluations, the surveyors highlighted the lack of gender representation among phone survey participants as it was mostly men who answered calls and were then interviewed. The surveyors also mentioned that eliciting responses to open-ended questions and confirming treatment compliance from every member of the household was challenging during phone based CES. These observations were confirmed

**Funding:** The DeWorm3 Project is funded by a grant from the Bill & Melinda Gates Foundation (OPP1129535, PI JLW). The funders had no role in study design, data collection, analysis, decision to publish, or manuscript preparation.

**Competing interests:** The authors have declared that no competing interests exist.

by analysing the survey participation data which showed women's participation in CES was significantly lower in phone-based CES (66%) compared to in-person CES (94%) ($Z = -22.38$; $p<0.01$) and that a significantly higher proportion of households provided proxy responses in phone-based CES (51%) compared to in-person CES (21%) ($Z = 20.23$; $p<0.01$).

## Conclusions

The phone-based CES may be a viable option to evaluate treatment coverage but issues such as participation bias, gender inclusion, and quality of responses will need to be addressed to optimize this methodology.

## Author summary

The DeWorm3 Project is a community cluster-randomized trial being conducted in Benin, India, and Malawi to test the feasibility of interrupting STH transmission by six rounds of biannual community-wide MDA (cMDA). As recommended by the World Health Organization, coverage evaluation surveys (CES) among 2,000 households were carried out in the DeWorm3 Project to validate reported treatment coverage of cMDA. During the COVID-19 lockdown in April 2020, the fifth round of CES was conducted using mobile phones instead of in-person survey. In phone-based CES, only 56% of sampled households were reached compared to 89% during the in-person CES. This was due to phone numbers being wrongly recorded, or calls being unanswered leading to a higher number of households that had to be sampled in order to achieve the sample size of 2,000 households in phone-based CES compared in-person CES (3,600 and 2,352 respectively). Focus group discussions conducted with the interviewers showed that phone-based CES took less time to complete than in-person CES; mostly men responded to the interview call; and eliciting responses to open-ended questions and confirming treatment compliance from every member of the household was challenging. These observations were also confirmed by analysing the survey participation data that showed less participation by women and more proxy responses in the phone based CES. Collecting phone numbers of women and avoiding open-ended questions may improve the women's participation and quality of responses.

## Introduction

India accounts for 21% of the 1.45 billion soil-transmitted helminths (STH) infections globally; including *Ascaris lumbricoides*, *Ancylostoma duodenale*, *Necator americanus*, and *Trichuris trichiura* [1]. In 2015, the Ministry of Health & Family Welfare (MOHFW), Government of India (GOI) launched the school-based National Deworming Day (NDD) program, wherein an estimated 240 million children between the ages of 1–19 years are dewormed biannually [2]. The World Health Organization (WHO) recommends coverage evaluation surveys (CES) to validate reported treatment coverage of mass drug administration (MDA) programs for neglected tropical diseases (NTD), like lymphatic filariasis (LF), onchocerciasis, schistosomiasis, STH, and trachoma [3]. In addition to estimating MDA coverage, CES helps to identify errors in reported coverage estimates and elicit reasons for non-participation that, if addressed, can help improve

coverage in future rounds [3]. The WHO recommends that CES be implemented at the community level even for school-based MDA programs to ensure that all out-of-school children, are also accounted for. The WHO also recommends that CES should be conducted by individuals not directly engaged in MDA delivery to reduce reporting bias.

The DeWorm3 Project is a community cluster-randomized trial being conducted in Benin, India, and Malawi, testing the feasibility of interrupting STH transmission by community-wide MDA (cMDA) [4,5]. In India, the trial's intervention phase launched in 2017, included six rounds of cMDAs (biannual) starting from 2018 to 2020, conducted immediately following the GOI's school-based biannual NDD program (the standard-of-care). The eligible population, between the ages of 1–99 years, for receiving STH treatment in cMDAs was estimated from the annual census of the study area. Within a week of completing each cMDA, in-person household CES were conducted to validate treatment coverage.

Following the fifth round of cMDA in March 2020, house visits for in-person CES could not be conducted in-person due to the nationwide lockdown announced by the GOI during the COVID-19 pandemic [6]. Therefore, the DeWorm3 team in India adopted an alternate strategy of collecting the fifth round of CES data via mobile phone voice calls (phone-based CES). Specific concerns related to telephone-based interview methods include sample representativeness of the population, response rates, and data quality [7]. To our knowledge, there is no available evidence regarding the feasibility and best practices of implementing CES using mobile phones. The purpose of this study is to describe the experiences of implementing a CES via mobile phones, including opportunities, challenges, and best practices compared to the standard in-person CES implementation.

## Methods

### Ethics statement

The Institutional Review Board (IRB) of Christian Medical College (CMC), Vellore (10392 [INTERVEN]), and the Human Subjects Division at the University of Washington (STUDY00000180) approved the DeWorm3 trial in India. The trial is registered at ClinicalTrials.gov (NCT03014167). The CMC IRB approved this study on phone-based CES as an amendment (IRB–A9, November 24, 2021, dated December 02, 2021). Written informed consent was obtained from all who participated in the focus group discussions and the in-person CES.

### Study location and setting

The DeWorm3 study in India is based in two sub-sites of the state of Tamil Nadu: the Timiri block in Ranipet district (formerly Vellore district) and Jawadhu Hills block in Tiruvannamalai district. A baseline census of the study area conducted from December 2017 to February 2018 enumerated a population of 140,932 individuals residing in 36,536 households. The population and socio-demographic details of the study area are described in detail elsewhere [8]. Briefly, the Timiri block is a rural area situated in the plains, whereas the Jawadhu Hills block is a tribal community located 762 metres above sea level. The project area is randomised into twenty intervention and twenty control clusters, 32 clusters in Timiri and 8 in Jawadhu Hills, ranging in population size from 3,179–4,561 per cluster [8]. During the baseline census, data regarding household socio-demographic characteristics, water, sanitation, and hygiene access, household address, contact phone number, and Global Positioning System (GPS) coordinates were collected and subsequently updated during annual census revisits. Community drug distributors (CDDs) were trained by the DeWorm3 study team to make house visits and distribute the treatment drug, Albendazole (200 mg for 1–2 years of age and 400 mg for all above two years of age). The study field workers accompanied the CDDs to directly observe the treatment, and

record individual and household-level cMDA delivery data on an electronic treatment register developed for the DeWorm3 trial [9]. Phone ownership details extracted from the latest census data of the two study sub-sites was mapped using the IDW tool embedded in ArcGIS (Version 10.3.1 for Desktop, ESRI, Redlands, California, United States of America).

## Coverage evaluation surveys

Fifty households per cluster were randomly selected to participate in each round of the CES (2,000 households total) [3]. The CES includes questions for each household member about their participation in deworming programs and reasons for non-participation. Information about the treatment history of children less than five years was collected from their primary caregivers. The DeWorm3 CES adapted the standardized WHO CES tool, with a mix of closed-ended questions and a small number of open-ended questions (S1 File) [3]. The survey questionnaire was translated into Tamil, the local language. To avoid any bias, the field workers and supervisors collected CES data from the clusters they were not assigned for carrying out cMDA. Written consent for the in-person CES was obtained from any adult member of the household present during administration. CES responses were recorded on a smartphone using the SurveyCTO mobile data collection application (Dobility, Inc; Cambridge, MA, and Ahmedabad, India) [4]. In April 2020, during the first COVID-19 national lockdown, phone-based CES was carried out instead of the in-person survey (Fig 1). The phone numbers provided by households during the annual census were used to contact randomly sampled households and administer the CES using the same questionnaire used for in-person surveys. Verbal consent for the phone-based CES was obtained from any adult member of the household. During the phone-based CES, two mobile phones were used by each officer, one for calling and another for recording the data on SurveyCTO. Here we analysed household sampling and household members participation in the CES.

## Study design and data collection

Eight CES implementers, six from Timiri and two from Jawadhu hills sub-site, who had experience in implementing both in-person and phone-based CES were recruited. The CES implementers of the two study subsites were invited to participate in a focus group discussion (FGD) conducted at their respective sub-sites. At both subsites, all the CES implementers agreed to participate in the FGD and provided written informed consent prior to starting the discussion. Two social scientists facilitated the interviews using a semi-structured question guide (S2 File). The question guide included questions about the advantages and disadvantages of conducting the CES using mobile phones compared to in-person and challenges related to eliciting response to questions on each platform. The FGDs were conducted in June 2020 when the pandemic lockdown following the first COVID-19 wave in India was relaxed. The discussions were facilitated in Tamil, the local language most comfortable to the survey implementers and audio recorded.

## Data analysis

The FGDs were transcribed verbatim and translated into English. The transcripts were coded by two primary coders independently using *a priori* thematic code list in ATLAS.ti 8.0 with a third coder as the arbitrator in case of disagreement between primary coders. The quantitative data was analysed using STATA 16.0 software (StataCorp, TX, USA), and Z test was used to test the difference between the two proportions.

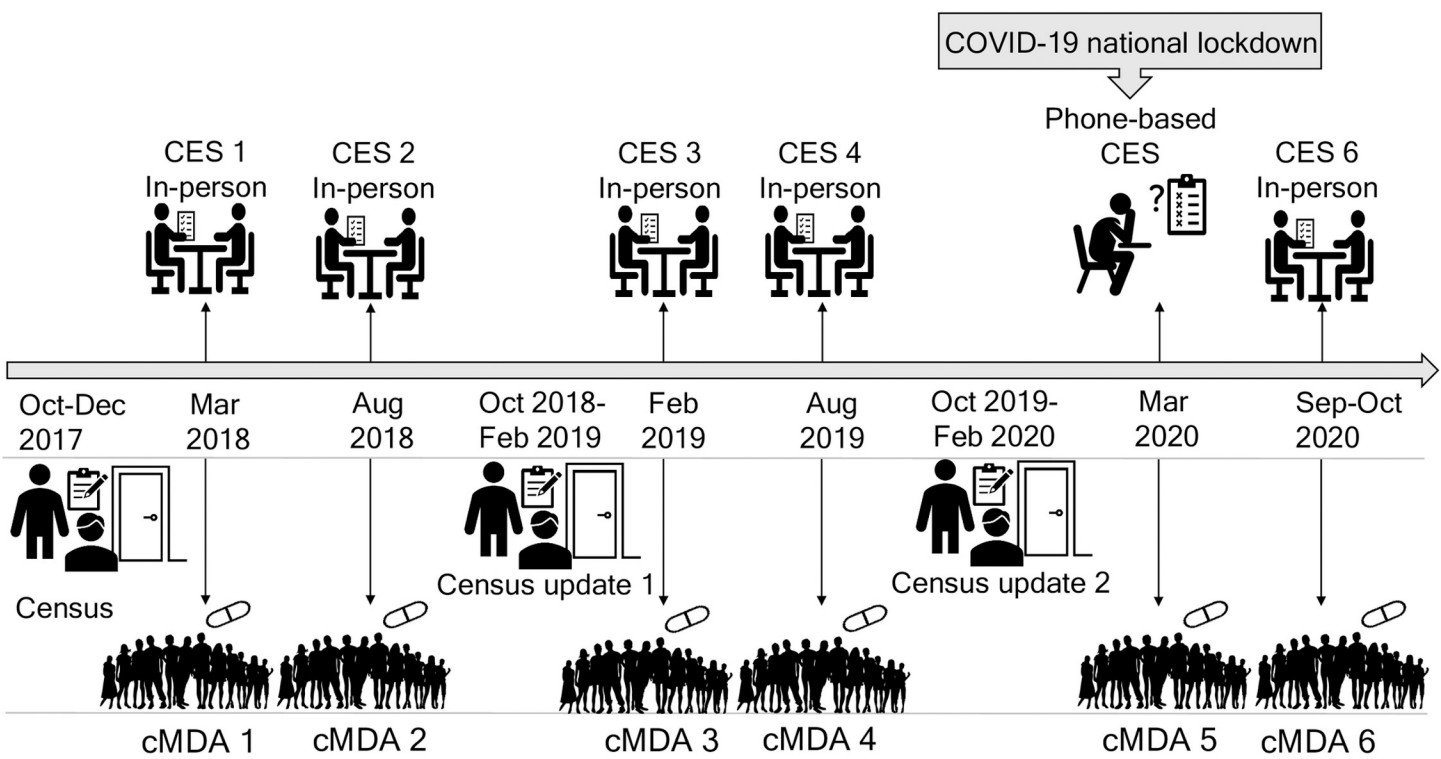

**Fig 1. Implementation timeline of census, cMDA, in-person CES and phone-based CES.**

## Results

Out of the eight survey implementers who had experience in administering both the in-person CES and phone-based CES, seven were men and one was a woman. They had an average of eight years of experience in public health programs or research, 3.5 years of experience in conducting the DeWorm3 trial, and were involved in all four rounds of in-person CES prior to the fifth round of phone-based CES. Primary themes from the FGDs included contacting households for the interviews, travelling to conduct the survey, building rapport, consenting for interview, achieving sample size, implementers' confidence of interview quality, time taken to interview, and gender of respondents. Their overall experience of conducting phone-based CES is summarised in Fig 2.

### Contacting households to participate was more challenging during phone-based CES than in-person CES

Challenges with contacting households in phone-based CES included poor network connectivity requiring multiple calls to connect, phone numbers recorded during census being incorrect, and the phone being continuously switched off. Some households had also given their neighbour's phone numbers, which made it difficult to talk to members of the sampled households immediately, and some of the selected households did not have a phone number. When the households sampled and participation was analyzed, in the phone-based CES, only 56% of sampled households were reached compared to 89% during the in-person CES. (Fig 3).

**Fig 2. Experience of conducting CES using mobile phones.**

In addition, most respondents preferred to speak during evening hours between 6–8 PM when they were at home, unengaged and free to speak and felt other times were inconvenient. The survey implementers felt that households not contactable over the phone could have been reached with an in-person survey. However, they noted that occasionally locating houses in more densely populated peri-urban areas was also a challenge despite having GPS coordinates, because the coordinates often identified the street but not the exact house. The implementers discussed as,

*"In XX cluster, we could not reach most households through the phone because they did not have good phone network connectivity. In face to face, we could go to the clusters and inter- view even if there is no network."* [Implementer #1, Jawadhu Hills]

*"In a village, it was enough if we just told the name of the head of the household, people would tell us where that person's house is. It was difficult to find houses in the town area, . . . it could have been a rented house, and they could have migrated. . . 5–10% of the houses cannot be located with GPS."* [Implementer #6, Timiri]

Even though there was no difference in the percentage of households with mobile phones between Timiri (85.9%) and Jawadhu Hills (85.6%), mapping household mobile phone owner- ship showed that access to mobile phones varied across study site due to topography. (Fig 4).

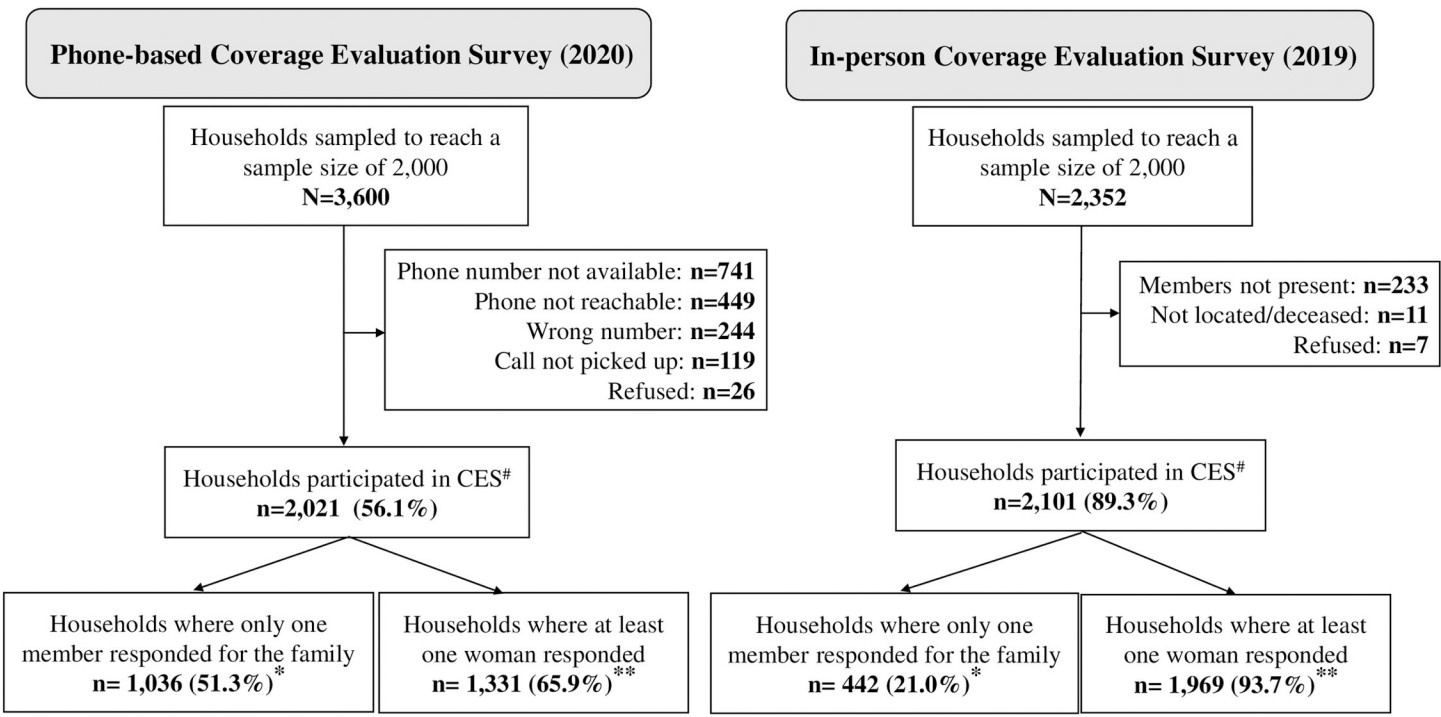

# CES: Coverage Evaluation Survey
* Z = 20.23, p-value <0.01; **Z = -22.38, p-value <0.01

**Fig 3. Household sampling and participation in phone-based CES compared to in-person CES.**

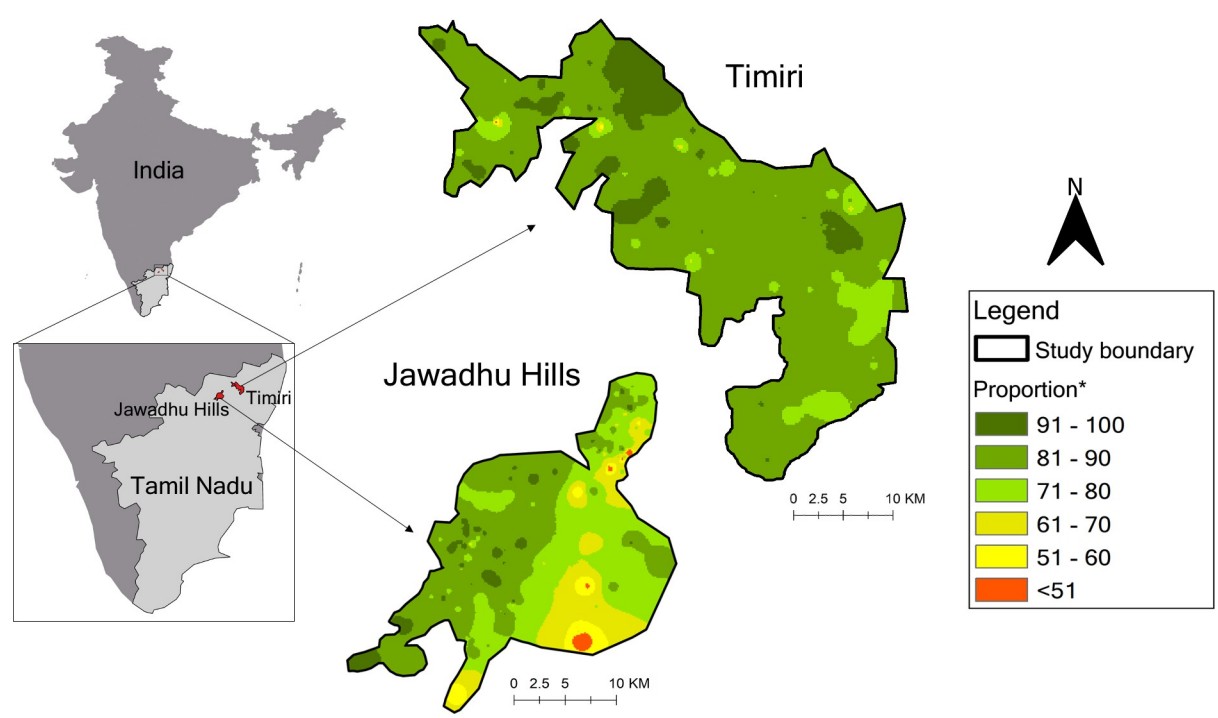

**Fig 4. Mobile phone ownership per 100 households by sub-sites, India.**

### Travelling to hard-to-reach terrain was minimised and travel time saved in phone-based CES

During in-person CES, the survey implementers sometimes needed to conduct two or three house visits for completing the questionnaire, particularly when a house was locked or if no adult was available for informed consent and interview. Phone-based CES eliminated the need for travel time, particularly for sampled households in Jawadhu Hills, where the terrain is hilly and road conditions are poor. As one of the survey implementers said,

*"In XX cluster, there is a village called YY, which had a sampled household. There is no road to go to that village, and there are full of stones, we have to walk for 20 minutes to reach that village. . .. We had to return without doing the interview when no adult was available to give written consent, which was a disadvantage in the in-person interview."* [Implementer #2, Jawadhu Hills]

### Building rapport was easier in the in-person CES than phone-based CES

Rapport was more easily established during the in-person CES interviews because the respondents recognised the survey implementers by their project uniform. Although the survey implementers explained and introduced themselves during phone-based CES interviews, the respondents did not warm up to the caller as easily. The survey implementers said,

*"On the phone, they would ask questions like, 'who is speaking, why are you speaking, how did you get my mobile number'. We had to explain the DeWorm3 project. . . it took more time."* [Implementer #2, Jawadhu Hills]

*"In a phone interview, we had to ask them politely for the interview like, 'due to corona we are not able to come to you in person, and that is why we are calling over the phone, so, please do not misunderstand us'. . .. Some people responded well. . . for some, we had to explain patiently to make them understand."* [Implementer #3, Timiri]

### Consenting was quicker in phone-based CES than for in-person CES

Particularly in the rural areas, the study implementers perceived fatigue among many respondents in providing written consent as the DeWorm3 trial protocol required written consent for several rounds of activities such as annual census, STH prevalence surveys, and CES. The study implementers also mentioned that the respondents faced the inconvenience of having to clean up their hands in the midst of carrying out household or other work before signing the consent form. Obtaining the verbal consent required in the phone-based CES protocol was, therefore, easier than obtaining written consent required for the in-person survey. However, in some tribal households, obtaining verbal consent over phone was challenging as the respondents had difficulty in recognising the caller. They said,

*"There were difficulties in getting the consent signed because we went to them many times in a year to get consent––for the census, for MDA and all that. . . . . .. we give them one copy of the consent, and we will take one copy. This will take around 10 minutes. . .. When we take signatures for consent, again and again, they ask us, "why are you asking for our consent again and*

*again'. Some of them hesitated to sign the consent. That difficulty was reduced over the phone because we got verbal consent."* [Implementer #1, Timiri]

*"When we went for the in-person interview, they immediately knew that we are people from the same locality. Everyone knew. . . They would welcome us and sign the consent readily. We had to explain a lot for a phone interview. . . If they know who is calling, it is easy; otherwise, tough to get consent over the phone. . . It will take more than 10 minutes just to explain. It is good to get consent in person."* [Implementer #2, Jawadhu Hills]

## A higher number of households had to be sampled in phone-based CES to achieve the sample size

The survey implementers reported that in the in-person CES, they were able to locate all the 50 sampled households. In case households were found to be locked even after three house visits then households from the second list of 20 sampled households were approached. Rarely, a third list of 20 sampled households was required to achieve the sample size in some clusters. In phone-based CES, third list of sample households was required to achieve the sample size in both the study sub-sites. Participants discussed the challenges with sampling via the phone-based survey by saying,

*"In MDA4, when we did in-person interviews, we completed more than 35 houses from stage-1 sampling list, and we would complete the rest with stage-2 sampling list.. . . Because many phone numbers were wrong and not reachable [during MDA5], we went to stage-3 sampled households."* [Implementer #1, Timiri]

*"We could reach 35 households out of the list of 50 households we got the first time. . .. In the in-person survey, we would have seen all 50 households. In- phone survey, we used second and third list of 20 households. . . 90 households to interview 50 households."* [Implementer #2, Jawadhu Hills]

The household sampling analysis also showed that a higher number of households (3,600) were sampled to achieve a sample size of 2,000 households in phone-based CES compared to 2,352 households in in-person CES (Fig 3).

## Implementers perceived lower confidence of response to questions in phone-based CES than in in-person CES

The CES questionnaire had both closed-ended and open-ended questions that needed further probing. For example, 'Why did you eat the tablet or not eat the tablet?', 'What did you like or not like about the STH cMDA program?', and 'Who gave you the tablet?'. Survey implementers reported that eliciting responses to these questions was more challenging over the phone compared to in-person interviews, as it was not always clear whether the respondent understood the question or not. It was particularly more challenging to elicit a response to such questions in the control clusters than in the intervention clusters. During in-person CES administration, the interviewer and the respondent could see and interpret each other's expressions and, accordingly, questions were repeated if needed based upon body language cues. The respondents would often become annoyed, particularly those in control clusters, when the question was repeated or probed over the phone for better clarity of responses like, 'Who gave you the tablet'. One of the implementers said,

*"In the intervention cluster, they clearly said that someone from the DeWorm3 project gave the tablet because they would have recognised them by their uniform but in the control cluster, they did not know who gave it. We had to ask to identify who gave the deworming tablet—a government nurse or ASHA, or DeWorm3 staff. We had to ask if the tablet was given using a spoon because, in DeWorm3, we give with a spoon."* [Implementer #2, Jawadhu Hills]

*"We can get good data in the face-to-face interview. When they answered, we could make out if they understood correctly or not and give the response accordingly or not. In the phone interview, we take only what they tell."* [Implementer #5, Timiri]

### Individual-level treatment compliance was more challenging to ascertain over the phone, as implementers often could not speak with every household member

The CES protocol included collecting responses directly from each household member above five years of age about their individual participation in the cMDA. In phone-based CES, collecting data from all family members was perceived to be somewhat feasible if the family size was small. This may be due to the ease of passing the phone from person-to-person when there were fewer individuals. In the case of extended and joint families, respondents were less willing to pass the phone to every member of the household and preferred to answer on behalf of all family members. During the in-person CES, it was possible to meet all family members and collect the information directly, even if they were engaged in other routine activities.

*"Households with four members. . . gave the phone easily for others. . .. If there are more members like 7–10, . . . only three of them talked. . .when the respondents were proxies for other members, we cannot say it was 100% accurate. . .. We had to explain that we would ask the same questions to other members and nothing else. . . Some people co-operated well. . .. In a face-to-face interview, we could ask everyone. . . even when they were watching TV, milking cows or cooking."* [Implementer #4, Timiri]

*"In the last MDA round, when we did coverage survey (in-person), someone initially said that he ate the tablet, but when we tried to confirm by asking the tablet's colour, he said he did not eat. So, initially, it was wrong information, and we could confirm in a face-to-face interview. . . So, there will be an opportunity to get an answer from every person directly."* [Implementer #1, Jawadhu Hills]

In confirming treatment compliance from family members, significantly higher proportion of households gave proxy response in phone-based CES (51%) than in-person CES (21%) (Z = 20.23; p<0.01) (Fig 3).

### Duration of interviewing a household was perceived to be shorter in phone-based CES than in-person CES

Completing the CES for a household over the phone took less time because respondents were often unwilling to pass the phone to their family members to get confirmation about consumption of deworming tablets, and instead confirmed consumption of deworming tablets on behalf of all family members (proxy response). In-person CES administration took longer because the interviewers spoke to all available family members. The participants reported,

*"For some people, it took time to explain over the phone. It would take more than 10 minutes just to explain. . .otherwise, we could finish the interview in 15 minutes. When we went for a face-to-face interview, it would take 20–40 minutes to complete a house because getting consent signed also took time."* [Implementer #1, Jawadhu Hills]

*"In a face-to-face interview, we could talk to all the family members and ask them many questions so took time. . .. They will spend a lot of time and talk in detail."* [Implementer #6, Timiri]

### Men were the primary respondents during phone-based CES compared to the in-person CES

The survey implementers perceived that more women were interviewed during the in-person survey as they were available at home during survey hours. In contrast, women only occasionally picked up the phone and responded to the interview in phone-based survey because men typically carry the household mobile phone. Men were often reluctant to pass the phone to women and adolescent girls of their households to answer CES questions or confirm their participation in treatment. The implementers explained as,

*"When we asked them (men) to give the phone to their daughters, if they (daughters) were 17 or 18 years old, they did not give and said something like, 'all of us took the tablet, why are you going and asking them'. . . they would say that their daughter-in-law is busy so to complete it (the interview) like this. . .. Many hesitated to give the phone"* [Implementer #2, Timiri]

*". . . mostly men picked up the phone, and they answered all the questions . . . when we wanted to ask the same from his wife, or children above 15 years or 18 years, teenage children, they do not accept. . .. Even after explaining, they would still say, 'I will answer'."* [Implementer #2, Jawadhu Hills]

Analysis of household participation showed that women's participation in CES was significantly lower in phone-based CES (66%) compared to in-person (94%) CES (Z = -22.38; p<0.01) (Fig 3).

### Discussion

Coverage surveys remain as an important component of STH treatment programs and are critical for monitoring, evaluation, and future planning to optimize programs. The lockdown in India due to COVID-19 in April 2020 offered a unique opportunity to explore CES using a phone call-based protocol. Overall, the phone-based CES was perceived to take less time to conduct and saved travel time, particularly in the hilly terrain areas. Consenting individuals was easier using the phone-based approach in the rural areas but no so in the tribal areas.

Phone-based survey requires high mobile phone penetration in the community, but this remains challenging particularly amongst groups with the lowest socioeconomic status [10–13]. Mobile phone ownership is 19 time lower in lowest socio-economic households than the highest socio-economic households [14]. However, in India, mobile phone subscription rates have shown significant increases with 11 million new subscribers added in just six months from July-December 2021, totalling over one billion with half the subscribers located in rural areas [15,16]. In the state of Tamil Nadu where the DeWorm3 Project is implemented, the

National Family Health Survey 5 (NFHS-5), showed a high household mobile phone owner-ship rate of 96% in urban areas and 90% in rural areas [17]. However, during our phone-based CES, we found that many households could not be contacted due to lack of a phone number or having provided a wrong phone number. The probability of entering wrong phone numbers could possibly be minimised during data collection and entry in the field by requesting the respondents to repeat the phone numbers and recording them at two separate instances with a logic check embedded in the electronic data entry form to flag and restrict progress to the next question if there is a discrepancy between both entries. Additionally, with increasing mobile phone subscription rates in India, some households may have more than one mobile phone and so alternate phone numbers may also be recorded during the census. It is important to note that calls may not have connected because mobile phone subscriptions may not have been renewed. Also, the mobile phone ownership in interior villages where the network con-nectivity is poor could be lower.

Most participants in the phone-based CES were men and the survey implementers observed reluctance among men to pass the phone to other household members, particularly women, adolescent girls, and the elderly, to confirm treatment compliance. Men may not serve as accu-rate proxy respondents in phone-based CES as they are less likely know who all in their house-hold had been treated. Under-representation of rural women was also observed in Afghanistan and Ethiopia when demographic surveys were conducted using mobile phones [13]. Analysis of meta-data from seven Indian surveys on livelihood and nutrition demon-strated that in rural households passing the phone to other household members to respond varied from about 8–11% across surveys, and that men carrying the mobile phone reduced a woman's likelihood to respond [18]. This study also demonstrated that women participation in the survey increased when they engaged more women survey implementers to conduct the interviews. The NFHS-4 and 5 conducted during 2015–16 and 2019–21, respectively showed an increase in mobile phone ownership among women from 62% to 69% among the rural women and 50% to 55% among scheduled caste women (most socially disadvantaged group) [17, 19]. With more women owning mobile phones, collecting their phone numbers has been shown to be feasible as the GOI also recommends collecting mothers' phone numbers in the maternal and child health program and COVID-19 vaccination programs [20,21]. Communi-ties could be informed during community sensitisation campaigns for various public health interventions if phone call surveys are planned, and women's phone numbers could also be collected during the program census or household visits.

The surveyors perceived confidence in the responses elicited to survey questions was lower in the phone-based CES compared to in-person CES. The WHO recommends that all house-hold members should be interviewed during a CES [3]. The survey implementers reported that they were only rarely able to interview all family members, particularly in households with large family sizes during the phone-based CES and so they were not confident about the validity of data collected given that responses were provided by one family member on behalf of all other members in the household. Coverage surveys following MDA for other NTD such as LF, onchocerciasis, schistosomiasis, STH, and trachoma programs from Malawi, Burkina Faso, and Uganda showed 1.7 times higher rate of drug consumption in proxy response compared to self-reported data [22]. Therefore, the reliability of proxy responses in coverage surveys needs to be examined, and ways to reduce proxy responses need to be explored [22]. Overreporting of drug consumption is expected where data are collected by proxy, even when the CES was conducted within one week of cMDA which is the best practice to implement CES [23].

The survey implementers also reported that it was challenging to obtain in-depth responses to open-ended questions during phone-based CES. A study from Portugal showed that non-responses for 'yes or no' questions in a phone call survey was lower than for open-ended

questions [24]. Therefore, the CES questionnaire designed to be administered by phone call may need to have more closed-ended questions to elicit a short response like 'yes' or 'no' or 'don't know' or have a shorter questionnaire focusing on the key questions of treatment offered and whether they consumed the tablet. The implementers experienced less resistance to obtaining informed consent during phone interviews than in the in-person interviews due to the time saved in obtaining a written consent. A review of telesurveys also recommended phone interviews in case of non-response in an in-person interview [25]. In our context, the response to a phone call was better in the evening when the respondents were at home. Similar observations were made in a study from Portugal where calling respondents when they are engaged in other tasks led to rushed responses [24]. Identification of the interviewer and rapport building are important aspects of interviewing, and this is particularly important in a phone-based survey when the respondent does not know who is calling. As the attention span of the respondent would be limited on a phone call, it may be worthwhile to train the interviewers to use a brief structured introduction of themselves. In addition, building community members' awareness about phone survey, prior to the program, may also help identify the caller and build better rapport for interviewing over the phone.

The implementers of the CES correctly observed that a higher number of households had to be contacted for phone-based CES due to higher rates of non-response in order to achieve the sample size compared to in-person CES. A literature review of telephone-based surveys has demonstrated coverage bias with underrepresentation of both older and less educated participants in these surveys [25–27]. Other secondary modes of data collection, including in-person, web-based, or email surveys, in addition to a phone-based survey as the primary mode, have been suggested to overcome sampling bias and increase the response rate [28].

Our experience of phone-based CES was based on conducting CES following cMDA for STH as part of a large-scale trial in rural and tribal areas in Tamil Nadu. This experience may not be generalisable to phone-based surveys in other geographies and may vary depending on the relevance and sensitivity of the information collected. Evidence from Lebanon indicates high concordance of health behaviour data like smoking, alcohol, diabetes, and hypertension, collected using mobile phones and in-person interviews [29]. On the contrary, a study including sensitive questions about well-being of individuals including loneliness and happiness and their financial situation indicated higher validity for phone call-based responses compared to in-person surveys [30]. Phone-based surveys are being recognised as a cost-effective method to collect real-time data compared to the traditional in-person surveys [31]. Though it was beyond the scope of this study, conducting a comparative costing analysis of phone-based and in-person surveys would have provided additional insights for decision making and planning such surveys.

## Conclusion

The use of phone-based CES appeared to be a feasible alternative when in-person surveys were not possible due to COVID-19 related lockdowns in India. However, this study found a number of challenges with phone-based CES methods. First, non-response rates were high during phone-based CES. Second, there was a clear gender bias in response, with mostly men responding to the calls. Third, obtaining responses to open-ended questions was challenging. Finally, interviewing every household member for CES, as suggested by WHO, was not feasible. Prior household sensitization about the importance of collecting phone numbers for program-related information; collecting alternate phone numbers of households, particularly women; confirming the correctness of phone numbers while recording during a census; and redesigning the CES questionnaire to avoid open-ended questions are some promising ways to

overcome the challenges identified. Using a combination of phone and in-person interviews in CES would increase reach of sampled households and save travel time.

## Supporting information

**S1 File. Coverage Evaluation Survey Questionnaire and Qualitative Guideline.**
(DOCX)

**S2 File. STROBE Statement.**
(DOC)

## Acknowledgments

The authors wish to thank all the participants of this study, communities and community leaders, national NTD program staff and other local, regional and national partners who have participated or supported the DeWorm3 study. We also would like to acknowledge the work of all members of the DeWorm3 study teams and affiliated institutions. We give special thanks to the field supervisors of the DeWorm3 study, India site, for their willingness to discuss their experiences of conducting coverage evaluation surveys using mobile phones and Uma Maheshwari for her support in transcribing the qualitative audio files.

## Author Contributions

**Conceptualization:** Kumudha Aruldas, Rohan Michael Ramesh, Sitara Swarna Rao Ajjampur.

**Data curation:** Angelin Titus, Jabaselvi Johnson, Malvika Saxena, Gideon John Israel.

**Formal analysis:** Kumudha Aruldas, Rohan Michael Ramesh, William E. Oswald, Gideon John Israel, Katherine Halliday.

**Funding acquisition:** Judd L. Walson.

**Investigation:** Kumudha Aruldas, Rohan Michael Ramesh, Angelin Titus.

**Methodology:** Kumudha Aruldas, Rohan Michael Ramesh.

**Project administration:** Sitara Swarna Rao Ajjampur.

**Resources:** Judd L. Walson, Sitara Swarna Rao Ajjampur.

**Software:** Jabaselvi Johnson, Malvika Saxena, Arianna Rubin Means.

**Supervision:** Kumudha Aruldas, Rohan Michael Ramesh.

**Visualization:** Kumudha Aruldas, Rohan Michael Ramesh, William E. Oswald, Venkateshprabhu Janagaraj.

**Writing – original draft:** Kumudha Aruldas, Rohan Michael Ramesh.

**Writing – review & editing:** Kumudha Aruldas, Rohan Michael Ramesh, William E. Oswald, Venkateshprabhu Janagaraj, Angelin Titus, Jabaselvi Johnson, Malvika Saxena, Gideon John Israel, Katherine Halliday, Judd L. Walson, Arianna Rubin Means, Sitara Swarna Rao Ajjampur.

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
