## [Decision Letter · Decision Letter 0]

10 Jun 2023

Dear Prof. Ajjampur,

Thank you very much for submitting your manuscript "Evaluating STH program coverage remotely - Experiences from the DeWorm3 study in India" for consideration at PLOS Neglected Tropical Diseases. As with all papers reviewed by the journal, your manuscript was reviewed by members of the editorial board and by several independent reviewers. In light of the reviews (below this email), we would like to invite the resubmission of a significantly-revised version that takes into account the reviewers' comments. 

We cannot make any decision about publication until we have seen the revised manuscript and your response to the reviewers' comments. Your revised manuscript is also likely to be sent to reviewers for further evaluation.

Sincerely,

Uwem Friday Ekpo, PhD

Academic Editor

Francesca Tamarozzi

Section Editor

Reviewer's Responses to Questions

**Key Review Criteria Required for Acceptance?**

**Methods**

-Are the objectives of the study clearly articulated with a clear testable hypothesis stated?

-Is the study design appropriate to address the stated objectives?

-Is the population clearly described and appropriate for the hypothesis being tested?

-Is the sample size sufficient to ensure adequate power to address the hypothesis being tested?

-Were correct statistical analysis used to support conclusions?

-Are there concerns about ethical or regulatory requirements being met?

Reviewer #1: see summary and general comment

Reviewer #2: This study describes the experiences of implementing a coverage evaluation surveys (CES) via mobile phones, including challenges and best practices compared to standard in-person CES implementation. Authors suggested some advices for using mobile phone-based CES as alternative tools to validate the treatment coverage of community mass drug administration when access to households was restricted. 

The objectives of the study are clearly articulated with a clear testable hypothesis stated, the study design is appropriate to address the stated objectives and the population is clearly described and appropriate for the hypothesis being tested. The sample size is also sufficient to ensure adequate power to address the hypothesis being tested.

However, the flow diagram is luck and will be interesting to clarify the sample aspects. Also the quantitative aspects needed to be added.

**Results**

-Does the analysis presented match the analysis plan?

-Are the results clearly and completely presented?

-Are the figures (Tables, Images) of sufficient quality for clarity?

Reviewer #1: see summary and general comment

Reviewer #2: The analysis presented match the analysis plan and the results are clearly and completely presented. However authors needed to include the quantitative aspects.

**Conclusions**

-Are the conclusions supported by the data presented?

-Are the limitations of analysis clearly described?

-Do the authors discuss how these data can be helpful to advance our understanding of the topic under study?

-Is public health relevance addressed?

Reviewer #1: see summary and general comment

Reviewer #2: The conclusions are clear and supported by the data presented

**Editorial and Data Presentation Modifications?**

Reviewer #1: (No Response)

Reviewer #2: This study describes the experiences of implementing a coverage evaluation surveys (CES) via mobile phones, including challenges and best practices compared to standard in-person CES implementation. Authors suggested some advices for using mobile phone-based CES as alternative tools to validate the treatment coverage of community mass drug administration when access to households was restricted. To improve the quality of responses and optimize phone-based CES authors suggested to confirming phone numbers while recording them, to collecting additional household phone numbers including those of women, and avoiding open-ended questions during the census step. This is certainly and interesting alternative coverage survey method and will need to be published. However, the following major concerns have emerged after review of the manuscript and will need to give more details/precisions:

This study was to be approached in two phases: Quantitative and Qualitative.

Quantitatively, we were hoping to see benchmarking of standard in-person CES previous used and phone-based CES, which will allow us to analyze the effectiveness of several parameters targeted in this study. What is the number of households called per cluster? for how many households was the call successful/contacting? Consenting? how many of repondents? … In this case a flow diagram of the selection of households would be necessary.

At the qualitative level: it is mentioned in Study design and data collection that "eight CES implementers, six from Timiri and two from Jawadhu hills sub-site, had experience of implementing both in-person and phone-based CES. All were invited to participate in a focus group discussion (FGD) conducted at each of the two sub-sites with two social scientists facilitated the interviews using a semi-structured question guide (S1file-Annex 2)". We understand that only two focus groups were carried out (1FGD in Timiri and 1FGD in Jawadhu). We understand also that the two CES implementers of Jawadhu participated in the FGD of Timiri and that the 6 CES implementers of Timiri also participated in the FGD of Jawadhu with the same questionnaire. Why didn't you think about creating the FGD of each sub-site with the CES implementers of each sub-site? Please clarify.

**Summary and General Comments**

Reviewer #1: The article is well written and the subject is very interesting, however in my opinion focus only on the logistical aspects of the phone-based CES provided by the implementers and do not provide any information about the most interesting parts: (1) the reliability of the results of the phone-based CES compared with the ones of the in person CES (2) the differential cost between phone-based CES and the in person CES.

Sincerely I do not think is possible to evaluate a method without knowing its performances, all the aspects analyzed by the authors (connection difficulties, gender bias, difficulties with open questions…) are of secondary relevance and should be interpreted on the light of capacity of the method to provide reliable data and save cost.

I suggest the authors to add in the results section of the MS (1) a report of the coverage results obtained by phone-based CES and an evaluation of its reliability, (2)an evaluation of the cost of the phone-based CES and the in person CES; If the reduction in cost is significant maybe an enlargement of the sample size to compensate the lower number of replies can be justified, but without a thorough evaluation of the performances of the methodology any conclusion on it has in my opinion limited value.

Reviewer #2: Please find as above

PLOS authors have the option to publish the peer review history of their article (what does this mean?). If published, this will include your full peer review and any attached files.

Reviewer #1: No

Reviewer #2: No

Figure Files:

Data Requirements:

Please note that, as a condition of publication, PLOS' data policy requires that you make available all data used to draw the conclusions outlined in your manuscript. Data must be deposited in an appropriate repository, included within the body of the manuscript, or uploaded as supporting information. This includes all numerical values that were used to generate graphs, histograms etc.. For an example see here: http://www.plosbiology.org/article/info:doi%2F10.1371%2Fjournal.pbio.1001908#s5.
---

## [Decision Letter · Decision Letter 1]

9 Sep 2023

Dear Prof. Ajjampur,

Thank you very much for submitting your manuscript "Remote evaluation of STH program coverage: Experiences from the DeWorm3 study, India" for consideration at PLOS Neglected Tropical Diseases. As with all papers reviewed by the journal, your manuscript was reviewed by members of the editorial board and by several independent reviewers. In light of the reviews (below this email), we would like to invite the resubmission of a significantly-revised version that takes into account the reviewers' comments. 

Editors Comments

Dear Author(s),

Thank you for the submission of your revised manuscript, which was re-evaluated by the previous reviewers. One reviewer in particular raised major concerns about the lack of cost analysis and validation of the reiiabilit of the phone-based interviews. The editors acknowledged that the inclusion of programme cost and tool validity is outside the scope of the study. However, we believed that adding quantitative anlyses to this manuscript and to apprise critically the results in the discussion would improve the information gathered on this topic and be of benefit to readers of PLoS NTD beside your intention to publish a separate assessment of the tool validity. 

Some examples of further analyses (of course not exhaustive of what should be added):

- % of cases where only one family member replied for everyone; 

- % increase in household contacts to be done compared to face-interviews;

- Another example, you stated that the fact that only the family head generally replied for other members, which results in a bias towards falsely higher report of drug intake compared to face-to-face interviews of all family members. So, was there any statistically significant difference in the intake report between the previous face-to-face interview rounds before-covid and the results obtained with the phone interviews?)

We cannot make any decision about publication until we have seen the revised manuscript and your response to the reviewers' comments. Your revised manuscript is also likely to be sent to reviewers for further evaluation.

Sincerely,

Uwem Friday Ekpo, PhD

Academic Editor

Francesca Tamarozzi

Section Editor

Editors Comments

Dear Author(s),

Thank you for the submission of your revised manuscript, which was re-evaluated by the previous reviewers. The editors agreed that the inclusion of programme cost and tool validity is outside the scope of the study. However, we believed that adding more quantitative analysis will enrich your manuscript. For example, you stated that the fact that only the family head generally replied for other members, which results in a bias towards falsely higher report of drug intake compared to face-to-face interviews of all family members. So, was there any statistically significant difference in the intake report between the previous face-to-face interview rounds before-covid and the results obtained with the phone interviews?

Therefore, adding some quantitative anlyses to this manuscript (e.g. % of cases where only one family member replied for everyone; % increase in household contacts to be done compared to face-interviews...) would improve the information gathered on this topic and be of benefits to readers of PLoS NTD beside your intention to publish a separate assessment of the tool validity.

Reviewer's Responses to Questions

**Key Review Criteria Required for Acceptance?**

**Methods**

-Are the objectives of the study clearly articulated with a clear testable hypothesis stated?

-Is the study design appropriate to address the stated objectives?

-Is the population clearly described and appropriate for the hypothesis being tested?

-Is the sample size sufficient to ensure adequate power to address the hypothesis being tested?

-Were correct statistical analysis used to support conclusions?

-Are there concerns about ethical or regulatory requirements being met?

Reviewer #1: (No Response)

Reviewer #2: This study suggest the phone-based overage evaluation survey CES as a viable option to evaluate treatment coverage when in-person coverage evaluation surveys was issue when access to households was restricted as it is the case during the COVID-19 pandemic. The objectives of the study are clearly articulated with a clear testable hypothesis stated and the the study design is appropriated to address the stated objectives. Globally the method is fine but should be improved.

**Results**

-Does the analysis presented match the analysis plan?

-Are the results clearly and completely presented?

-Are the figures (Tables, Images) of sufficient quality for clarity?

Reviewer #1: (No Response)

Reviewer #2: Interestingly, the results are clearly and completly presented.

**Conclusions**

-Are the conclusions supported by the data presented?

-Are the limitations of analysis clearly described?

-Do the authors discuss how these data can be helpful to advance our understanding of the topic under study?

-Is public health relevance addressed?

Reviewer #1: (No Response)

Reviewer #2: Yes the conclusions suggested are supported by the data presented and authors discuss how the data can be helpful to improve the new method suggested quality. Authors point out a number of challenges with phone-based CES methods: high non-response rates, gender bias in response and, the challenge to obtaining responses to open-ended questions.

**Editorial and Data Presentation Modifications?**

Reviewer #1: (No Response)

Reviewer #2: No comments

**Summary and General Comments**

Reviewer #1: The Author did not made any substantial change to the paper: The paper in the present version address only the opinions of the users without any information on performance and cost of the tools.

since the authors are mentioning that the tool performances are addressed in a different paper, I suggest the authors to join the two papers in a single one with all the relevant information.

this will allow the reader to judge about the applicability of the tool.

In the present form the MS do not merit publication in my opinion as the opinion of the user is not a sufficient element to evaluate the usefulness of the proposed tool.

Reviewer #2: Globally the reviewers suggestions have been duly incorporated and response have been given accordingly. Therfore this manuscript could be accepted for publication in Plos NTD

PLOS authors have the option to publish the peer review history of their article (what does this mean?). If published, this will include your full peer review and any attached files.

Reviewer #1: No

Reviewer #2: Yes: Ibikounlé M

Figure Files:

Data Requirements:

Please note that, as a condition of publication, PLOS' data policy requires that you make available all data used to draw the conclusions outlined in your manuscript. Data must be deposited in an appropriate repository, included within the body of the manuscript, or uploaded as supporting information. This includes all numerical values that were used to generate graphs, histograms etc.. For an example see here: http://www.plosbiology.org/article/info:doi%2F10.1371%2Fjournal.pbio.1001908#s5.
---

## [Editor Report · Decision Letter 2]

24 Oct 2023

Dear Prof. Ajjampur,

We are pleased to inform you that your manuscript 'Remote evaluation of STH program coverage: Experiences from the DeWorm3 study, India' has been provisionally accepted for publication in PLOS Neglected Tropical Diseases.

Best regards,

Uwem Friday Ekpo, PhD

Academic Editor

Francesca Tamarozzi

Section Editor

---

## [Editor Report · Acceptance letter]

30 Oct 2023

Dear Prof. Ajjampur,

We are delighted to inform you that your manuscript, "Remote evaluation of STH program coverage: Experiences from the DeWorm3 study, India," has been formally accepted for publication in PLOS Neglected Tropical Diseases.

Best regards,

Shaden Kamhawi

co-Editor-in-Chief

Paul Brindley

co-Editor-in-Chief
